

# GLOBALLY ORTHOGONAL (GO) INITIALIZATION FOR MLP

## HANWEN WANG, ISABELLE CRAWFORD-ENG, PARIS PERDIKARIS

## Applied Mathematics & Computational Science

## MOTIVATION

- Multi-layer Perceptrons (MLPs) often suffer from frequency bias and vanishing gradients.
- The widely used Glorot initialization scheme [1] initializes the weights to $\mathcal{N}(0, c/\sqrt{d_{in}})$, $c = 1$, so that the linearly transformed hidden outputs preserve their second moment.
- However, non-linear activation functions may trigger collapsed hidden outputs, especially for deep MLPs at initialization. (e.g. Fig. 1, for $c = 1$).
- The choice of $c$ for each hidden layer at initialization matters.

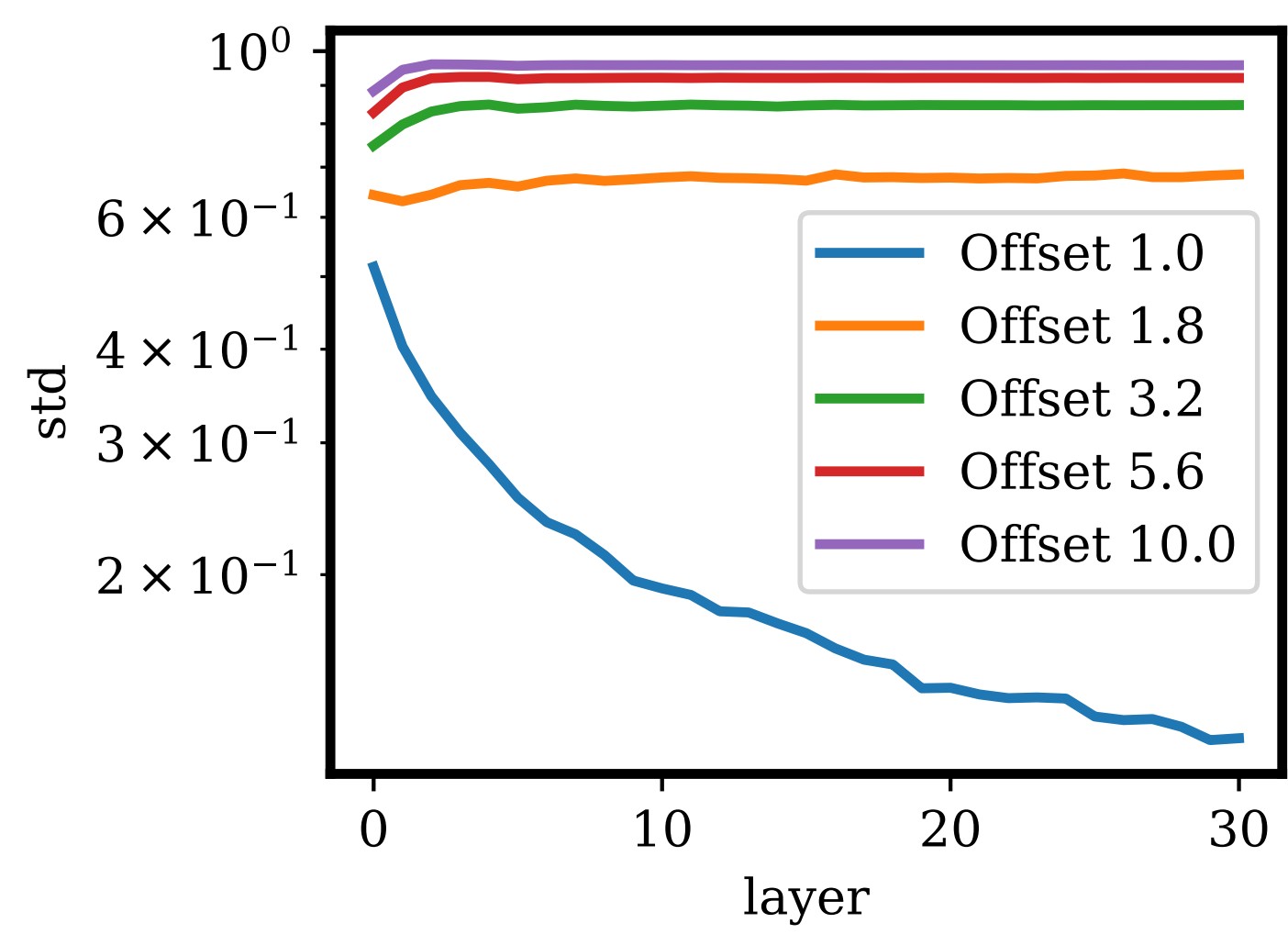

**Figure 1:** Evolution of hidden layer output standard deviation, given inputs with a standard normal distribution.

## FLOW OF VARIANCE

- Consider a linear layer $Y = XW$. Entries of $W$ are independent and identically distributed (i.i.d) with zero mean and standard deviation $c/\sqrt{d_{in}}$.
- Given the the square root of input second moment $s_i$, and activation $f_i$, the square root of the post-activation second moment can be obtained as

$$s_{i+1} = \sqrt{\mathbb{E}(f_i(cs_i\mathcal{N}(0,1))^2)},$$

for an infinitely wide hidden layer.
- We simulate the above system for common activation functions and initializations [1, 2, 3].
- Our simulations (see Fig. 2) confirm our analysis for existing initializations.
- Stable fixed points at 0 imply collapsed outputs especially for deeper MLPs.

See https://openreview.net/forum?id=KkMGjzTsXM for the workshop submission.

## FLOW OF VARIANCE

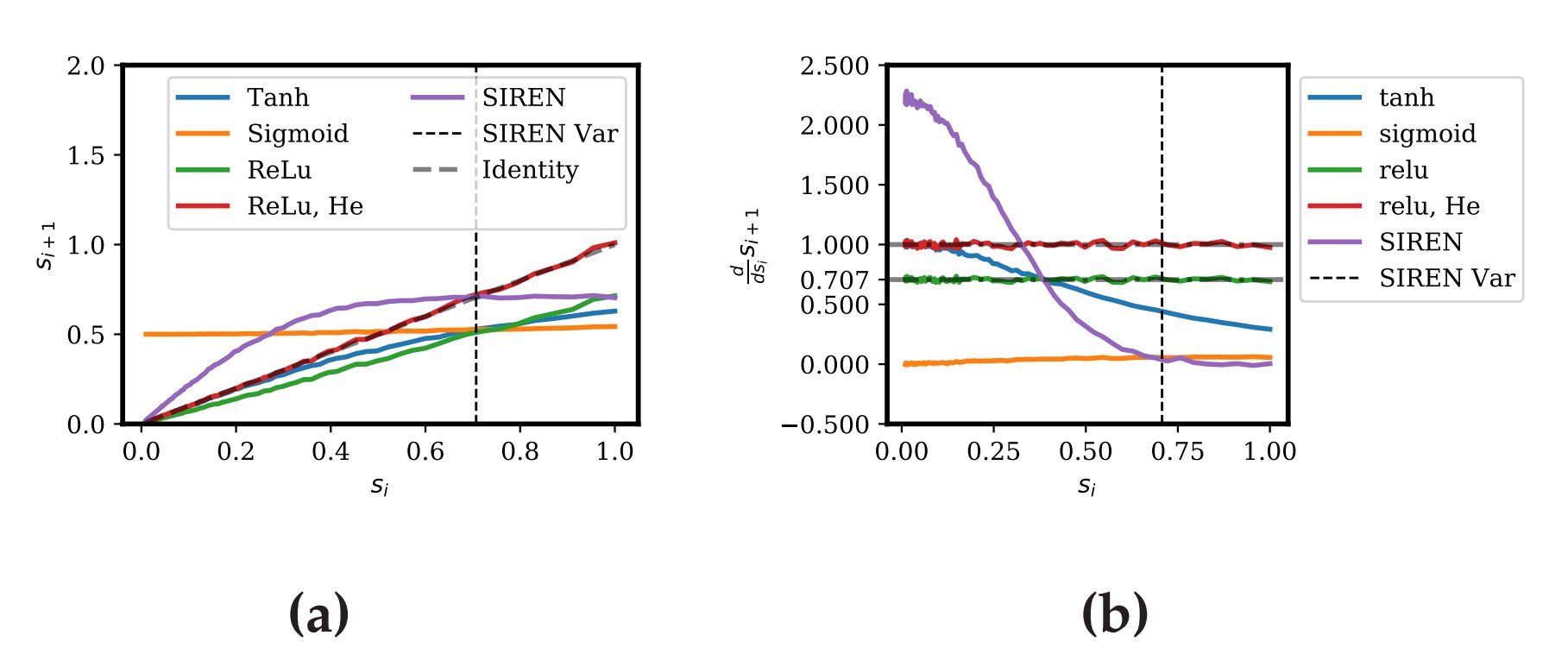

**Figure 2:** A simulation of the dynamical system in the previous section. (Left) The iterative mapping functions for Glorot, He, and SIREN initializations. (Right) The derivative of the iterative mapping functions shown in the left.

## GO INITIALIZATION OF MLPS

- Use orthogonality in the last layer features as a measure to quantify the expressivity of MLPs in the data domain.
- Calculate the similarities (cosines of angles) of the last layer hidden features.

$$\cos\theta^{i,j} := \frac{\langle \boldsymbol{g}_w^i, \boldsymbol{g}_w^j \rangle}{||\boldsymbol{g}_w^i||_2 ||\boldsymbol{g}_w^j||_2}.$$

- Transform the similarities between features to a regularized loss objective as

$$\min_{\boldsymbol{c},\boldsymbol{b}} -\frac{\sum_{i\neq j}\log(1-(\cos\theta^{i,j})^2)}{d(d-1)} + \lambda||\boldsymbol{c}||_2^2.$$

- Minimizing this loss at initialization leads to MLP weights that prevent collapsed outputs by penalizing similarities in the last layer features.

## NTK ANALYSIS

- The Neural Tangent Kernel (NTK) of MLP networks with inputs $x, y$ is defined as

$$\boldsymbol{K}(x,y) = \nabla_w f_w(x) \cdot \nabla_w f_w(y).$$

- Several studies [4, 5, 6] show that the NTK of infinite width MLPs converges to a deterministic kernel, and is invariant under gradient descent with infinitely small learning rate.
- Deep MLPs with tanh activations initialized using the s GO scheme tend to exhibit a localized NTK.
- This localized behavior discounts long-range correlations between distant data points, and hence promotes the fitting of local function behavior.
- Beneficial for learning functions with high frequency components.

## NEURAL TANGENT KERNEL (NTK) ANALYSIS

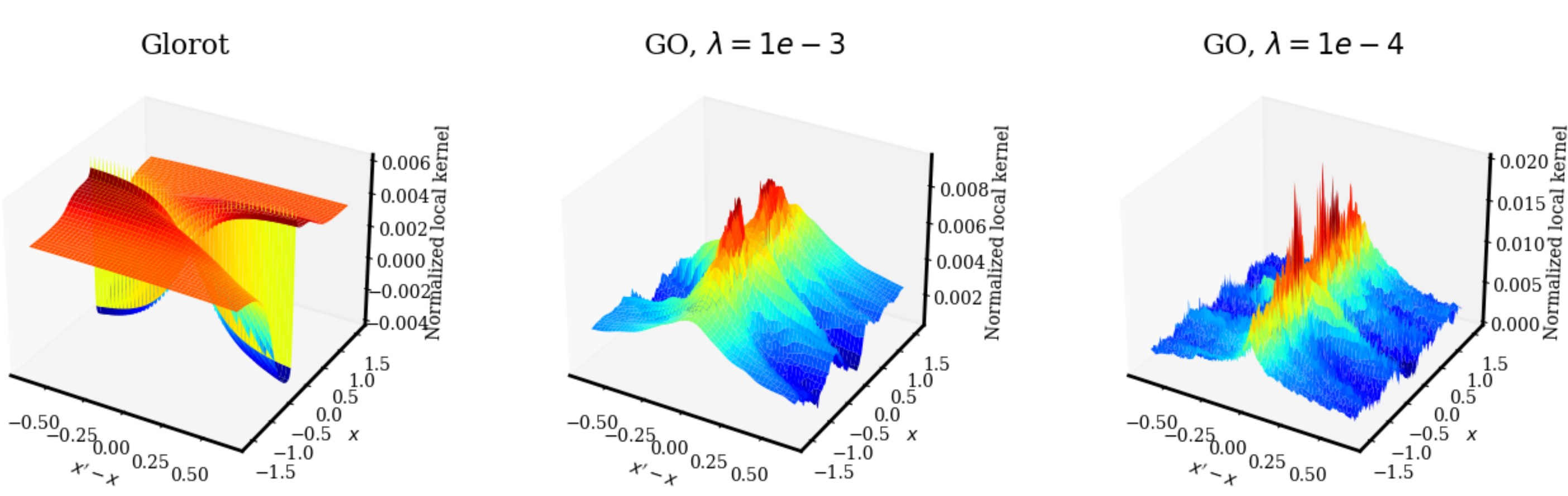

**Figure 3:** NTK in the data domain for MLP initialized with Glorot scheme (left), and the proposed GO scheme with different regularization strength (middle and right).

## NUMERICAL EXAMPLES

**Synthetic data:** Synthetic data-set from a highly oscillatory ground truth function. In different trials, data points are sampled denser to impose larger oscillations. Fig. 4 shows that scale-adjustment from GO initialization brings more robust fitting.

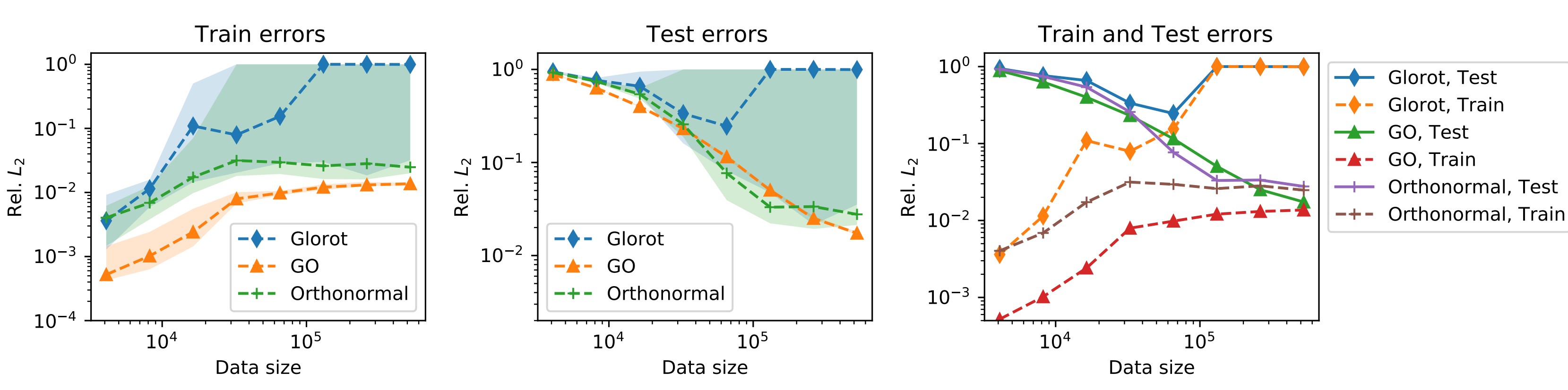

**Figure 4:** Train and test errors for comparison and contrast. (Left) Medium train errors and 80% CIs. (Middle) Medium test errors and 80% CIs. (Right) Train and test errors convergence.

**Image regression:** The ground truth RGB image contains rich details of a city landscape. A continuous representation of the image is recovered by fitting 25% of the pixels using an MLP with total variation regularization and tanh activations. An MLP with GO initialization is able to improve the SNR from 16.8 dB to 20.0 dB compared to the Glorot scheme.

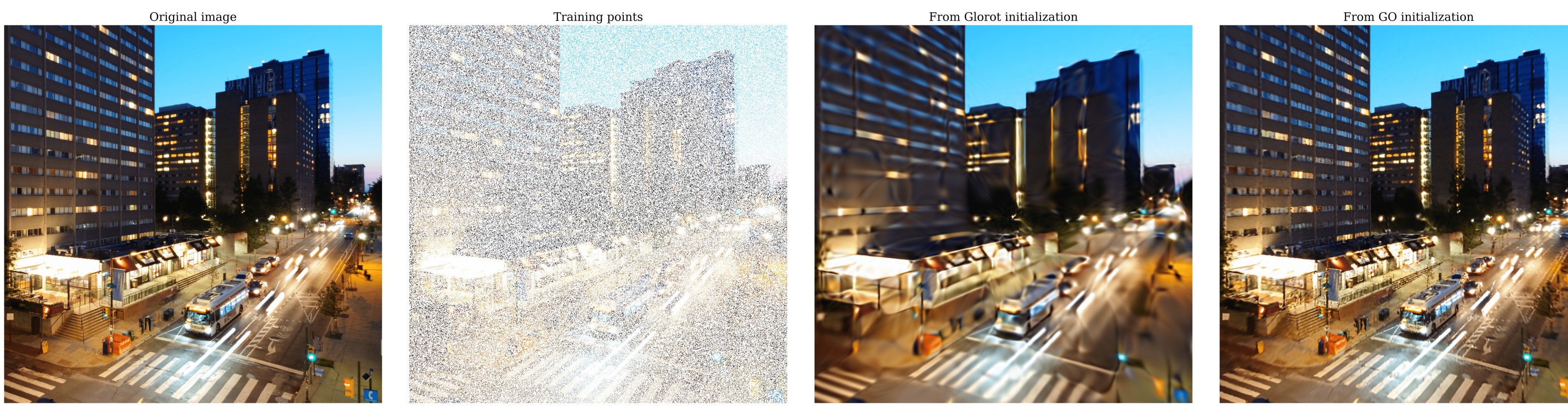

**Figure 5:** Image regression results. (From left to right) Original image, sampling mask, learning from Glorot initialization, learning from GO initialization.

[1] Xavier Glorot and Yoshua Bengio. Understanding the difficulty of training deep feedforward neural networks. In Yee Whye Teh and Mike Titterington, editors, *Proceedings of the Thirteenth International Conference on Artificial Intelligence and Statistics*, volume 9 of *Proceedings of Machine Learning Research*, pages 249–256, Chia Laguna Resort, Sardinia, Italy, 13–15 May 2010. PMLR.
[2] Kaiming He, Xiangyu Zhang, Shaoqing Ren, and Jian Sun. Delving Deep into Rectifiers: Surpassing Human-Level Performance on ImageNet Classification. In *2015 IEEE International Conference on Computer Vision (ICCV)*, pages 1026–1034, Santiago, Chile, December 2015. IEEE.
[3] Vincent Sitzmann, Julien NP Martel, Alexander W Bergman, David B Lindell, and Gordon Wetzstein. Implicit neural representations with periodic activation functions. *arXiv preprint arXiv:2006.09661*, 2020.
[4] Arthur Jacot, Franck Gabriel, and Clément Hongler. Neural tangent kernel: Convergence and generalization in neural networks, 2020.
[5] Sanjeev Arora, Simon S Du, Wei Hu, Zhiyuan Li, Russ R Salakhutdinov, and Ruosong Wang. On exact computation with an infinitely wide neural net. In H. Wallach, H. Larochelle, A. Beygelzimer, F. d'Alché-Buc, E. Fox, and R. Garnett, editors, *Advances in Neural Information Processing Systems*, volume 32. Curran Associates, Inc., 2019.
[6] Jaehoon Lee, Lechao Xiao, Samuel S Schoenholz, Yasaman Bahri, Roman Novak, Jascha Sohl-Dickstein, and Jeffrey Pennington. Wide neural networks of any depth evolve as linear models under gradient descent. *Journal of Statistical Mechanics: Theory and Experiment*, 2020(12):124002, Dec 2020.
