# OpenReview forum: "Enhancing the trainability and expressivity of deep MLPs with globally orthogonal initialization"
_NeurIPS.cc/2021/Workshop/DLDE — DLDE Workshop -- NeurIPS 2021 Poster_

### Official Review · Reviewer_Brgm · 2021-09-29
**This paper tries to tackle the problem of SGD based optimization algorithms using a different approach from a dynamical systems viewpoint.**

**Confidence:** 4

**Review:**

This paper addresses an important problem while training neural networks.

Pros:
1. This paper is written nicely with a clear motivation.
2. The authors back their ideas with a range of experiments.

Cons:
1. There is a lack of novelty in the method proposed by authors.
2. The figures are presented without much explanation. It is not clear what do we infer from those figures.
3. Comparison with previous approaches is lacking.
4. More experiments including ablation study would be better.

**Score:**

2: Borderline paper

---

> ### Author Response · Authors · 2021-10-22
> **Response**
>
> Due to the four page length limit of this submission, we have reluctantly removed comments and results that could have addressed the issues raised by the reviewer. We are planning to discuss some of these issues more extensively in a forthcoming longer version of our manuscript.

---

### Official Review · Reviewer_NNoa · 2021-10-11

**Confidence:** 3

**Review:**

- Summary

This paper proposes an initialization scheme for neural networks that aims to make the outputs of the last layer of the neural network have orthogonal values.

- Main review

I will start by admitting I am not thoroughly familiar with the literature on the diverse initialization schemes for neural networks, and so it is hard for me to judge the novelty of this paper in context of existing work.

It is also not clear to me that the ideas in the paper relate properly to the theme of this workshop, but I will leave this evaluation to the meta-reviewer.

This paper proposes a clear idea, namely initializing neural network parameters so that the outputs of the last layer are orthogonal (or close to that). This is performed by solving an optimization problem that also performs of a trade-off of such orthogonality and a regularization of the parameters.

It is not very clear why we should care about the orthogonality of the outputs of the initialized network. There is only a very brief high level discussion. A deeper discussion of why this matters would be important.

Moreover, the issue of frequency bias is mentioned repeatedly, yet it is not discussed why this orthogonalization procedure should be expected to fix the frequency bias.

The authors mention that previous methods have focused on performing initializations that orthogonalize the parameters of the network (not just the outputs). However, the experiments do not compare against these methods, (only against standard Xavier initialization) which would seem to also be an important comparison point.

It is not clear to me from the descriptions of the methods and the experiments how the cost of the optimization that is performed to optimize the network in this "global orthogonal" fashion compares to existing methods, the cost of training the network, etc. It is also not clear how this cost scales with network size.

A minor point is that the naming used in Figures 5 and 6 is confusing, as it does not match the names used in the text (Xavier/Glorot, GO/Scaled). Moreover, in Figure 5, the fact that the colors do not match between corresponding lines in the first two plots and the last one makes it hard to parse the results.

In line 101, the term "significantly" is used to express the difference between the evaluated methods. This term is usually reserved for a statistical test sense, but here seems to be used loosely (as the values from one method seem to be within 80% confidence bounds of the other).

Nevertheless, despite these issues, the paper presents some interesting results on the experiments performed. If some of the issues above are addressed it should make it good enough for acceptance.

**Score:**

2: Borderline paper

---

> ### Author Response · Authors · 2021-10-22
> **Response**
>
> We thank the referee for reading through our manuscripts and providing such a detailed and thoughtful review. Due to the four page length limit of this submission, we have reluctantly removed comments and results that could have addressed the issues raised by the reviewer. We will briefly present these results and comments here, as follows.
>
>
> Connection between this submission and the theme of this workshop.
>
> One of the target applications of the proposed initialization scheme is to mitigate frequency bias that prevents MLPs from learning of high frequency functions. This is crucial for popular frameworks such as Physics-Informed Neural Network, where MLPs are employed to approximate the solution of differential equations, albeit they often fail for cases where the solutions exhibit multi-scale features.
>
> Motivation to optimize the orthogonality.
>
> As shown in our discrete dynamical system modelling the information flow from one layer to the next, the conventional Glorot initialization scheme often leads to collapsed outputs at initialization, where the output vanishes to almost zero in the domain, if the domain is not properly normalized. Such collapse then requires prolonged training time to learn a target function. In this case, the orthogonality is low since the last layer hidden features are almost identically zero. However, our model suggests that with larger variance offset, the output may not vanish even for deep MLPs. And in this case the orthogonality of the last layer hidden features is indeed much greater. This observation motivates the use of orthogonality in the last layer hidden features as an objective to optimize the network’s parameters at initialization.
>
> Why the proposed initialization helps overcome the frequency bias.
>
> We use Neural Tangent Kernel (NTK) theory to empirically show that the proposed initialization scheme helps to prevent frequency bias. It has been proven that the NTK of an infinitely wide neural network under the gradient flow training dynamics is invariant, and that the result of regression with an infinitely wide neural network is equivalent to a kernel regression with Laplace kernel. We use the NTK to show that, empirically, the length parameter of the NTK with the proposed initialization scheme is shorter than that initialization with the Glorot scheme, and thus the result of the kernel regression will encourage the learning of high frequency components since the high frequency components are less likely to be averaged to zero.
>
> Comparison to other initialization schemes.
>
> We have included some additional comparisons in the revised manuscript, and we plan to include more in a longer version of our manuscript.
>
> No cost of training in the proposed initialization scheme.
>
> We have included a relevant discussion in the revised supplementary material.
>
> Phrasing and typos.
>
> We have carefully revised the manuscript and corrected typos and grammatical errors.

---

### Official Review · Reviewer_fsaq · 2021-10-12
**An interesting take on initialization**

**Confidence:** 5

**Review:**

The authors consider the question of weight initialization for MLPs (i.e., fully-connected deep neural networks) from the perspective of dynamical systems. Whereas traditional initialization schemes have primarily emphasized the stability of neural activations in very deep networks, here the authors propose a novel initialization scheme that emphasizes the orthogonality of neural features in the output layer of the network at initialization. They provide evidence that the scheme produces more complex outputs at initialization than standard methods, with a greater variety of localized features uniformly distributed throughout the domain. Among other things, they argue that this property makes it suitable for remedying the frequency bias normally responsible for the failure of MLPs trained on functions with multiscale or small-scale features.

In my opinion, the submission is interesting and worth including in the workshop. However, I think the authors can significantly improve this work by making better connections to related works, and by better motivating the significance/impact of their results for the DLDE/SciML commmunities more broadly.

The authors should certainly consider relating their work to the Box initialization scheme of Cyr et al. (https://arxiv.org/abs/1912.04862). This work offers a very similar perspective to neural network initialization, and also finds that the standard schemes used in mainstream deep learning are sub-optimal for use in the physical sciences. A side-by-side comparison of the two would be very interesting.

The authors may also be interested in the recently released Principles of Deep Learning Theory (https://arxiv.org/abs/2106.10165). That work provides a comprehensive model of network initialization schemes based on drawing weights and biases from i.i.d. Gaussian distributions. It may be interesting to compare the modelling in that work to Equation 2.1 in the current submission. Indeed, I think the derivation of Equation 2.1 would benefit from a bit more explanation, especially to clarify what approximations are being made. Although I may be missing something, the conclusion that tanh networks have two fixed points seems to contradict the results of chapters 4 and 5 of the Principles of Deep Learning Theory, which (I believe) argue that tanh activations lead to networks whose only fixed point (for the variance of network activations) is at zero (and, moreover, that this fixed point is stable). I am sure there are subtleties I am missing at the moment, but a more in-depth comparison of the two approaches would be interesting.

Another relevant work is this one: https://arxiv.org/abs/2008.09878. Equation 1 of that work can be contrasted directly with Eqn 3.2 of the current submission. Both losses are designed to encourage orthogonality of the features in a neural layer, although Singh et al. applied theirs to the first hidden layer, whereas the authors here have applied theirs to the output layer.

As the other reviewers have pointed out, it is unclear at first glance that this submission fits the themes of the workshop. Upon closer inspection, I think that it does indeed, but that the authors missed an opportunity to make this more clear. The frequency bias of MLPs is a major barrier to their use for representing the solutions to many interesting PDEs, so an improvement here is relevant to the DLDE community. Moreover, the orthogonality of features in the last layer is reasonably well motivated by the fact that other numerical methods seek linear combinations over orthogonal function spaces to represent solutions to PDEs; for instance, the finite element method works well for precisely this reason. Line 67 does not do justice to this motivation, which I feel is important for connecting the submission to interests of the broader DLDE community. Indeed, the work of Cyr et al. also explore this perspective, relating the neural network method of solving differential equations to adaptive basis methods (and extending this view to the LS/GD optimization scheme as well).

As the other reviewers have mentioned, many of the figures have legends/labels that do not match the wording in the text (e.g., 'Scaled' in Fig 5 is the GO method). Similarly, many of the hyperparameter settings and experimental details are relegated to the supplemental, but this is not clearly stated in the main body of the submission.

Nonetheless, the results in Figs 5 and 6 appear very impressive. In particular, the experiment in Fig 6 suggests this scheme may have applications beyond the physical sciences.

The authors comment at the beginning of the paper that they are motivated by the empirical fact that MLPs do not achieve their theoretical potential in practice. It would thus be interesting to related this work to available approximation theories for MLPs; see for instance https://arxiv.org/abs/1610.01145, https://arxiv.org/abs/1705.01714, and other works by the same authors. In particular, for the synthetic data examples used in this work it should be possible to predict the theoretically rate of convergence of MLPs versus network capacity. It is known that these rates are not normally achieve in practice (e.g., https://arxiv.org/abs/1910.12686); does this new method come closer to realizing the ideal rates?

In B.5, the authors briefly mention that, when comparing to the SIRENs method, they do not apply any heuristics in the initialization of the first layer. Are they referring to the use of a factor of 30 to increase the frequencies in the first layer in the SIRENs paper? Although it is only motivated loosely in the SIRENs paper, it is eminently reasonable and general approach to capturing finer length scales in the training domain, and the use of that factor is likely very important to the success of the SIRENs method. Thus, it seems unfair to omit it when comparing the new method to that one. Moreover, the authors' GO method is an initialization scheme, and as such can readily be combined with the sine activation function. A full comparison of GO with tanh, GO with sine, SIRENs without the factor of 30, and SIRENs with the factor of 30 might be helpful in disambiguating the relative contributions of the two initialization schemes and the two activation functions.

**Score:**

3: Good paper

---

> ### Author Response · Authors · 2021-10-22
> **Response**
>
> We thank the reviewer for reading through our manuscript and supplementary material, and providing such detailed and thoughtful feedback. Due to the four page length limit of this submission, we have reluctantly removed comments and results that could have addressed the issues raised by the reviewer. We will briefly present these results and comments here, as follows.
>
> Relating the proposed initialization scheme to the Box initialization scheme.
>
> Indeed the Box initialization also focuses on preserving the shapes of the hidden features from layer to layer. However, it is introduced for ReLU and other piece-wise linear activations while ours focus on the MLP with Tanh activation. The latter are better suited in cases where MLPs are used to parametrize the solution to differential equations (e.g physics-informed neural networks). In such cases one needs to differentiate the neural network outputs with respect to its inputs, and ReLU activations lead to discontinuous derivatives. Nevertheless have included a short discussion on Box initialization, and we are planning to provide a more extensive comparison in a longer version of our manuscript.
>
> Comparison to the result in Principles of Deep Learning Theory.
>
> Our model of MLP initialization is much simpler than that in the book Principles of Deep Learning Theory. While our fixed point is merely the value that the square root of the second moment of the hidden outputs converges to with certain variance offsets, the analysis in the book involves some kernel that is about the correlation between neurons for fixed input. We therefore believe that these definitions may differ from ours in our text.
>
> Comparison to the method of Gurpreet Singh at eq 3.2.
>
> Indeed, eq. 1 in that paper also considers the similarity between the hidden features. However, their objective is more direct than others, and the collapsing may still occur while the features are passing through the later layers. This is the primary reason that we apply our objective on the last layer features. And the computation of the gradient to that objective affects the parameters of all layers during training, whose cost could be much higher than ours that only includes two parameters per layer at initialization.
>
> Theoretical complexity analysis
>
> We must first admit that the theoretical complexity analysis, and the asymptotic trend of convergence of errors as the neural network grows wider and deeper, are hard to obtain due to the limit of the first order gradient based optimization methods. We will plan to present additional results on this in a longer version of our manuscript.
>
> More comparison to SIREN.
>
> We have included some additional comparisons to SIREN in the supplementary material as the referee has suggested.
>
> Phrasing and typos.
>
> We have carefully revised the manuscript and corrected typos and grammatical errors.

---

### Decision · Program_Chairs · 2021-10-16

**Decision:**

Accept (Poster)

**Comment:**

Reviewers were uncertain about this paper; in particular its relevance to the DLDE community.

Overall I am inclined to accept it, but this was a very borderline decision. I would echo the reviewers' sentiments that much stronger connections should be drawn to dynamical systems.